# Great Enhancement of Carbon Energy Storage through Narrow Pores and Hydrogen-Containing Functional Groups for Aqueous Zn-Ion Hybrid Supercapacitor

**DOI:** 10.3390/molecules24142589

**Published:** 2019-07-16

**Authors:** Chao Liu, Jian-Chun Wu, Haitao Zhou, Menghao Liu, Dong Zhang, Shilin Li, Hongquan Gao, Jianhong Yang

**Affiliations:** 1School of Material Science and Engineering, Jiangsu University, 301, Xuefu Road, Zhenjiang, Jiangsu 212013, China; 2Key Laboratory of Radiation Physics and Technology, Ministry of Education, Institute of Nuclear Science and Technology, Sichuan University, Chengdu 610064, China

**Keywords:** proton transfer mechanism, Zn-ions, functionalized activated carbon, PANI nanofibers, pH values

## Abstract

The proton transfer mechanism on the carbon cathode surface has been considered as an effective way to boost the electrochemical performance of Zn-ion hybrid supercapacitors (SCs) with both ionic liquid and organic electrolytes. However, cheaper, potentially safer, and more environmental friendly supercapacitor can be achieved by using aqueous electrolyte. Herein, we introduce the proton transfer mechanism into a Zn-ion hybrid supercapacitor with the ZnSO_4_ aqueous electrolyte and functionalized activated carbon cathode materials (FACs). We reveal both experimentally and theoretically an enhanced performance by controlling the micropores structure and hydrogen-containing functional groups (–OH and –NH functions) of the activated carbon materials. The Zn-ion SCs with FACs exhibit a high capacitance of 435 F g^−1^ and good stability with 89% capacity retention over 10,000 cycles. Moreover, the proton transfer effect can be further enhanced by introducing extra hydrogen ions in the electrolyte with low pH value. The highest capacitance of 544 F g^−1^ is obtained at pH = 3. The proton transfer process tends to take place preferentially on the hydroxyl-groups based on the density functional theory (DFT) calculation. The results would help to develop carbon materials for cheaper and safer Zn-ion hybrid SCs with higher energy.

## 1. Introduction

Supercapacitors (SCs) offer high specific power density and long life cycles [1,2]. However, the low energy density (full-cell energy density of 6–10 Wh L^−1^) limits the wide application of SCs. Meanwhile, the typical energy density for widely used lead-acid battery is 50–80 Wh L^−1^ [3]. So, improving energy density of the SCs has always been the hotspot in the energy storage field. Efforts have been devoted to neat ionic liquid (IL) electrolyte, which delivered high operating voltage range of 3–5 V [4,5]. However, compared to the commercial organic electrolyte, the IL is expensive and the resistance for ion transport is large due to the relatively strong cation–anion interactions within the electrolyte, which is harmful to the power of the device [6]. In addition, materials with intrinsic pseudocapacitive properties, such as MnO_2_ [7], Nb_2_O_5_ [8], conductive polymers [9], and costly RuO_2_ [10], usually have fast ion transport tunnels or surface reactions. However, most of these materials have limited electronic conductivity, short life, and complicated preparation process, limiting their application on the high power-type devices.

The development of hybrid capacitors further increases the energy density of SCs. For instance, the Li-ion hybrid SCs have achieved great success in the commercial application [11,12]. Alternatively, Na^+^ [13,14], Mg^2+^ [15], Zn^2+^ [16], K^+^ [17], etc. hybrid capacitors were also reported by researchers. Recently, Kang et al. [18] developed a safe, good rate capability, and long life Zn-ion hybrid SCs using the ZnSO_4_ aqueous electrolyte, where the Zn^2+^ ions were plated on or dissolved from the Zn foil, and the SO_4_^2−^ anions were adsorbed to or desorbed from the activated carbon. Such device delivered specific energy of 84 Wh kg^−1^ (based on the mass of carbon positive electrode materials). Commercial activated carbon (AC) with pore size of 1.2–2.7 nm was used for the positive electrode, which gave a technological potential for the improvement of energy density for such device by using optimized carbon materials. In fact, as early as 2006, the Y. Gogotsi’s group found the anomalous increase of carbon capacitance at pore sizes less than 1 nm, especially for the electrolyte with solvated ions [19]. This result was mainly due to the desolvation of the electrolyte ions entering narrow micropores, leading to a sieving effect at pore sizes below the solvated ions size. In addition to the narrow pores optimization work, the other method to improve the SCs performance is the pseudocapacitance [20,21]. As early as 2009, the François’s group [22] increased the SCs energy density in H_2_SO_4_ electrolyte using the seaweed biopolymer carbon with abundant surface functions resulted in redox reactions. Recently, Zhou et al. [23] found an enhancement of the Zn-ion SC capacitance through the pseudocapacitance induced by proton transfer in both the ionic liquid and organic electrolyte. Hence, the proton transfer mechanism should be also valid for the Zn-ion hybrid SC with aqueous electrolyte.

Here, a new method was developed to increase the capacitance by combining narrow pores and proton transfer mechanism of aqueous Zn-ion hybrid SCs with functionalized polyaniline (PANI)-derived activated carbon positive materials (FACs). A one-step KOH activation process [24] was used to achieve the optimized micropore-rich structure and generate rich hydrogen-containing functional groups. The aqueous Zn-ion hybrid SC delivered the maximum specific capacitance of 435 F g^−1^ (101 Wh kg^−1^) at a high power density of 14.1 kW kg^−1^ based on the mass of FACs. Moreover, the SC showed good stability with capacity retention of 89% for 10,000 cycles. In order to enhance the effect of proton transfer on the electrochemical performance, extra H^+^ was injected into the electrolyte by adjusting the pH value. The carbon nanosponges (CNSs) [4,5] with mesopores-rich structure and few functional groups and FACs after heat treatments were also prepared and systematically investigated for better comparison in this work.

## 2. Results and Discussion

### 2.1. Effect of the Pore Morphology

The morphologies of functionalized activated carbon materials (FACs activated at 700 °C without heat treatment, see Appendix A: Experimental section) are shown in Figure 1a,b. Unlike the carbon nanosponge materials (CNSs) with a pyrolysis step (Appendix A) [4,5], the FACs show a flat surface different CNSs sample with a fiber structure. The bridged and linked porous structure collapsed after direct (one-step) KOH activation of the PANI fibers, as same as the reported work by Wang et al. [25]. On the flat surface of FACs, micropores were homogeneously distributed, as shown in the TEM image (Figure 1b). The pore characteristics and pore size distributions (PSDs) of FACs, heat-treated FACs, and CNSs were verified by performing N_2_ (77.4 K) adsorption/desorption measurements, as shown in Figure 1c,d. All the FACs samples have lower adsorption amounts than the CNSs sample, especially within a range greater than 0.2 of relative pressure, demonstrating low developed porosity. The adsorption nearly keeps a constant value at the relative pressure >0.2, indicating that fewer mesopores and macropores exist in the FACs materials (Figure 1c), which are consistent with the SEM and TEM observations (Figure 1a,b). The pore size distributions (PSDs) (Figure 1d) of the six samples were calculated from the N_2_ isotherms by the nonlocal density functional theory (NLDFT, Tarazona) using a slit pore structure model [26]. All the FACs samples have high peaks at 0.6 nm compared to CNSs sample. The BET specific surface areas (SSA) and NLDFT pore volume of FACs without heat treatment are calculated to be 1968 m^2^ g^−1^ (with 93.1% Horvaih Kawazoe (HK) microporous SSA) and 0.82 cm^3^ g^−1^. The average pore size of the FACs is 1.85 nm. The CNSs sample shows SSA and pore size up to 2735 m^2^ g^−1^ and 2.52 nm, which is higher than other samples. The results of pores structures for these six samples are presented in Appendix A. It should be noted here that the FACs with low surface area and micropores-rich structure give much higher tapping density than the mesopores-rich carbon materials [24]. The compacted density of the FACs electrode can be as high as 0.5 g cm^−3^ after hot rolling, which is critical for the practical application on the energy storage device with higher volumetric density and process ability [27,28].

The effect of PSDs on the electrochemical performance was explored, PANI-based activated carbon materials with different NLDFT PSDs were prepared by changing the one-step activation temperature (650, 700, 750, 800, 850, and 900 °C). In addition, functionalized carbon nanosponge mateirals (FCNSs) [23] (as reported in our previous PANI-based carbon work by using an H_2_O/CO_2_ co-activation process) and CNSs [4,5] (prepared by using KOH activation with pre-pyrolysis step) are also used to compare with the FACs samples. All the calculated parameters of pores structure are also listed in Appendix A for better comparison. When the values of specific capacitance were normalized by SSA, the effect of PSDs, without the consideration of surface area, could be verified (Figure 2b). Figure 2b shows a trend of increasing normalized capacitance when the HK micropores SSA ratio increases. Especially, a sharp increase in capacitance can be observed when the ratio increases to higher than 80%. Similar phenomena have been reported by Chmiola et al. [19]. In this case, for the electrolyte with solvated sulfate anion, Wan et al. [29] proposed that the smallest hydrated sulfate anion clusters have a full solvation shell with 12 water molecules. The geometry size of the SO_4_^2−^ (H_2_O)_12_ cluster was calculated to be 0.78 nm using the density functional theory (DFT), as shown in Figure 2a (Supplementary computational details). However, a strong influence between the adjacent ions and the distance from the anion to the carbon charged surface should also be considered [30]. Thus, the effective size of the cluster near the electrode surface is ~1 nm before entering into the pores. If most of the carbon pores are in the size of ~0.6 nm (Figure 1d), the solvation shell is distorted as the ion is crowded into the narrower pore. The distortion gives less distance from the ion center to the carbon surface (smaller d value, as shown in Figure 2b), which leads to enhancement of the capacitance.

### 2.2. Effect of the Surface Composition

The effect of the temperatures of heat treatments on the surface composition of carbon materials was investigated by FTIR and X-ray photoelectron spectroscopy (XPS). First, the functions of the six samples were qualitatively examined by FTIR. Figure 3a shows the FTIR spectrum for the six samples. –OH and –NH stretching occurred at 3200–3600 cm^−1^ [31]. A strong band at 1030 cm^−1^ has been connected with the C–N stretching [32]. The relative peak strength at 605 cm^−1^ attributed to the −OH bending vibrations [33] reduces with the heat treatment temperature increasing, meaning the decrease of –OH contents after the heat treatments. The FTIR spectra show that the CNSs sample has fewer functions than the FACs samples. Then, the quantitative examination of surface functional groups by XPS was given in Figure 3b. The O1s relative intensities (~533 eV) decrease with the heat treatment temperatures. The N1s peaks (~400 eV) were barely detectable due to extremely low N contents in CNSs samples. However, it can be clearly observed in all the FACs samples spectra. As the temperature reaches 800 °C, the C at% increases slightly, the O at% decreases from 11.1 to 9.0%, and the N at% decreases from 4.0 to 2.9%. The sample of CNSs has the highest C at% of 91.2%, as listed in Appendix A. The XPS peaks were adopted to deconvolve the high-resolution C1s, O1s, N1s peaks. The deconvolution of the C1s, O1s, N1s spectra gave five, four, and three peaks, respectively [34]. Figure 3c show typical XPS results together with fitted O1s of six samples, and all the N1s results for the five FACs samples are given in Figure 3d. The corresponding group of each peak is marked on the schematic diagram of the FACs structure (Figure 3e). All the fitted C1s, O1s, N1s peaks for each sample are listed in Appendix A. The analyses of C1s, O1s, N1s peaks are summarized and shown in Appendix A.

With the temperature increasing, the C_I_ (graphitic carbon) increased, the C_II_ (C–OH) reduced, and C_III_ and C_IV_ (carbonyl functions) increased. It coincided well with O_I_, assigned to carbonyl oxygen, increased from 47.46 to 59.48% (Appendix A). Moreover, the O_II_ and O_III_ corresponding to the –OH groups decreased (Figure 3c). Meanwhile, the relative amount of N_III_ (the quaternary N) increased and that N_II_ (the nitrogen bonded to amine functions) decreased (Figure 3d) due to the increase of temperature. Based on the above surface characterization results, during the heat treatment, the elimination reactions should take place to reduce the hydrogen-containing groups, the C–OH, C–H, and C–NH reacted to form the C=O, quaternary N, and H_2_O.

Before the SCs performance testing works, the electrochemical mechanism was determined by using a three-electrode cell. Appendix A shows the cyclic voltammetry (CV) curve of the cell obtained by using the Zn foil as the working electrode and the FACs electrode as the counter electrode, and a tiny Zn metal wire as the reference electrode between potentials of −0.8 to 1 V versus Zn/Zn^2+^ at a scan rate of 5 mVs^−1^. In the cathodic process, the Zn^2+^ ions deposit on the Zn foil and the Zn_4_SO_4_(OH)_6_•5H_2_O precipitation may also form based on the study by Dong et al. [18]. In the anodic process, the Zn stripping happens and Zn_4_SO_4_(OH)_6_•5H_2_O precipitation re-dissolves into the electrolyte. Appendix A shows the CV curves of the 3-electrode cell obtained by switching the working electrode to the FACs cathode with the Zn foil electrode serving as the counter electrode. In the anodic process, the SO_4_^2−^ ions are adsorbed on the FACs electrode. The peak at right end side is tilted, meaning that the O_2_ evolution reaction may happen at the potential higher than 1.6 V vs. Zn/Zn^2+^. However, the side reaction at high voltage was eased when we tested the CV in the 2-electrode setup using coin cell due to the low potential of Zn anode (~ −0.2 V vs. Zn/Zn^2+^), as shown in Appendix A. In the cathodic process, the SO_4_^2−^ ions are desorbed from the FACs electrode.

Figure 4a presents the cyclic voltammetry (CV) curves for the FACs, heat-treated FACs, and CNSs samples at 5 mV s^−1^ by adopting three-electrode system. CNSs sample shows the almost rectangular shape of the CV curve, which is typical electric double-layer capacitors (EDLC) behavior. However, for the FACs sample, the distinct arc-shape can be observed during both the cathodic (0.8 V vs. Zn/Zn^2+^) and anodic (1.0 V vs. Zn/Zn^2+^) processes, meaning that some reversible redox reactions contribute to the capacitance of the electrode. The pseudocapacitance effect gradually reduced with the increased heat-treated temperature, and the CV curve showed an almost rectangular shape for the FACs800 sample. This phenomenon is consistent with our previously reported work [23], which can be well explained by the “proton transfer mechanism”. It should be noted here that the FACs samples are rich in nitrogen-containing functional groups, which are also considered in this work. Figure 4b lists the changes in functional group contents. For comparison, the average specific capacitances for each sample are given in a bars diagram with error-bars (Figure 4b). For each sample, more than four SCs were assembled and measured by the galvanostatic charge/discharge (GCD) method at 0.1 A g^−1^ (Appendix A). The change of discharge specific capacitance is consistent with the trend in the change of O_II_ + O_III_ and N_II_ contents, corresponding to the –OH and –NH functions, respectively. For the FACs with or without heat treatments, the micropores ratios are close to each other. Hence, the pores structure effect can be ignored and the surface chemistry effect is ascertained. The characterization results revealed the hydrogen-containing functional groups are the key factor on the electrochemical property of the FACs-based SCs with aqueous electrolyte. The redox reactions between the hydroxyl and amino groups on the graphene layer and the sulfate group in the electrolyte were studied by DFT, as shown in Figure 4c,d. During the chemisorption, the proton can transfer from the hydroxyl (Figure 4c) and amino groups (Figure 4d) to the sulfate group with the total energy reduce of 2.35 eV and 1.35 eV, respectively. The results may also indicate that the proton transfer process tends to take place preferentially on the hydroxyl groups. Furthermore, the amino groups will join the reaction following with the reactions on the hydroxyl groups. Based on the above calculation, the faradaic charge transfer reactions between the hydrogen-containing functional group and the sulfate group can be expressed as shown in Figure 4c,d. Both the initial adsorption states and the final relaxed states are also given in Figure 4c,d.

The FACs electrode in the SCs with aqueous electrolyte was further studied by the separation of the surface redox reaction and EDLC contribution from current response of CV test [35,36]. As discussed above, the proton transfer is attributed to a diffusion-controlled process (*i* ∝ *v*^1/2^), while the surface adsorption/desorption is resulted in the capacitive process (*i* ∝ *v*). Hence, the total current (i) of CV test can be expressed as [37]: *i* = *k*_1_*v* + *k*_2_*v*^1/2^,(1)

Capacitive behavior (*k*_1_*v*, shadow area) and diffusion control at a scan rate of 5 mV s^−1^ are shown in Figure 4e. Approximately 33% of the total current is contributed by the diffusion-controlled process at 5 mV s^−1^. The detailed calculations are given in Appendix A.

More interestingly, the proton transfer effect can be further enhanced by introducing extra hydrogen ions in the electrolyte. The pH values of electrolyte were adjusted to 3 and 5 by adding 1 M H_2_SO_4_ solution. The specific capacitances of the FACs samples with electrolyte pH values of 7, 5, and 3 are 435, 477, and 544 F g^−1^ at a low current density of 0.1 A g^−1^, respectively. This improvement was mainly attributed to the increase of the diffusion-controlled capacitance, resulting from the proton transportation. The contribution of faradaic reaction decreased as the sweep rate increased, which is coincident with the other works, as shown in Figure 4f. At the same sweep rate, the contribution ratio of diffusion-controlled process was decreased with the pH value. For instance, diffusion-controlled ratio was increased from 37 to 78% as the pH value changed from 7 to 3 at a low rate of 1 mV s^−1^, as shown in Figure 4f, meaning more H^+^ ions take part in the faradaic reaction based on the proton transfer mechanism.

### 2.3. Electrochemical Performance of FACs

Figure 5a presents the result of CV tests of the FACs-based (without heat treatment) SC over a wide range of scan rates from 1 to 50 mV s^−1^ using three-electrode cells in aqueous electrolyte. The big arc shapes at 0.8 V and 1 V vs. Zn/Zn^2+^ indicate the existence of pseudocapacitance, which is different from the EDLC rectangular shape. The GCD curves of the FACs-based SCs at different current densities based on the mass of FACs are shown in Figure 5b. The SCs delivered high specific capacitances of 435, 365, 280, 233, 220, 215, 214, and 198 F g^−1^ at 0.1, 0.2, 0.5, 1, 2, 5, 10, and 20 A g^−1^ (0.5 C, 1 C, 2.5 C, 5 C, 10 C, 25 C, 50 C, and 100 C, 1 C = 0.2 A g^−1^), respectively. The curves have distinct distortions, especially at low current densities (0.1–2 A g^−1^). The Coulombic efficiency is 85.3%, 94.3%, 97.9%, 99.4%, 100%, 100%, 100%, and 100% at the current density of 0.1, 0.2, 0.5, 1, 2, 5, 10, and 20 A g^−1^, respectively. Figure 5c shows the specific capacitances calculated from the galvanostatic discharge curves of the six samples. The initial capacitances of the FACs without and with heat treatments are 435, 408, 364, 313, and 266 F g^−1^ at 0.1 A g^−1^, respectively, which decreased with the heat-treated temperature (from 200 to 800 °C). At high current densities, the FACs has good rate performance and retention of 45% at 20 A g^−1^. The CNSs and the heat-treated samples at 200, 400, 600, 800 °C have the capacitance retention of 50%, 39%, 41%, 45%, and 44% at 20 A g^−1^, respectively.

The specific energy of the FACs-based Zn-ion hybrid SCs was estimated from the GCD discharge curve based on the mass of FACs. The highest specific energy of the FACs-based SC was 101 Wh kg^−1^ (0.1 A g^−1^, based on the weight of FACs). Moreover, the FACs offered 46 Wh kg^−1^ when the specific power was above 14,100 W kg^−1^. The Ragone plot of specific energy versus specific power is given in Figure 5d. In order to compare with our previous works, the volumetric energy density of the device was also estimated by the volume factor of the electrode material in a packed cell based on the compacted density and thickness ratio of the active material [38]. The estimated volumetric energy densities and other commercial energy storage devices were plotted in a Ragone plot (Appendix A). The energy density of FACs-based Zn-ion hybrid SCs (26.3 Wh L^−1^) was about 3–4 times higher than the commercial SCs based on organic electrolyte (3–6 Wh L^−1^) [1,39]. For practical application, an electrode with a thickness of 120 μm was also fabricated. The thick electrode with high loadings delivered a maximum specific energy density of 75 Wh kg^−1^, relatively low rate capability, and poor power performance. Considering the 10 μm stainless steel current collector and 20 μm Zn foil, the specific energy of the full devices was estimated to be 28 Wh kg^−1^ at low power density of 2 W kg^−1^, and 9 Wh kg^−1^ at 1180 W kg^−1^, which is still higher than the commercial SC, as shown in Figure 5d.

High electrochemical cycle stability is found for FACs at 2 A g^−1^ based on the weight of FACs in coin cell, as shown in Figure 5e. Excellent stability with 89% capacity retention was obtained over 10,000 cycles, indicating the SC has a good cycling performance. The results show that the pseudocapacitance introduced by the hydrogen-containing functions in the aqueous electrolyte is stable with cycling. However, a sudden capacity drop at around 3000 cycles and fluctuation of the Coulombic efficiency values are also shown in Figure 5e. These phenomena could be due to the continuous gas evolution. Coffee bag (soft-pack) cells for the Zn-ion hybrid SC were also made as our previous works [23], and the inflation of the bag was found after cycling. The gas evolution could be due to two reasons. The first one is the O_2_ evolution on the carbon cathode side during charge at high voltage. The functions on the carbon surface may also catalyze the oxygen evolution [40]. However, based on the CV results (Figure 4a), the CNSs with the highest surface area and least functions show the lowest O_2_ evolution over potential and highest peak at high potential region, meaning the specific surface area (SSA) is a key factor for the side reaction. Both the Coulombic efficiency and stability are improved, when the voltage window was fixed at the range of 0.3–1.6 V, as shown in Appendix A. The other reason is the H_2_ evolution on the Zn anode side. Especially, low pH value can decrease the H_2_ evolution over potential of the Zn anode. Appendix A shows the cycle stability of the Zn ion hybrid SC with electrolyte pH values of 3 and 5. The capacitance retentions are 92% and 88% after only 1000 cycles for pH 3 and 5, respectively, and the fluctuation of the Coulombic efficiency values was also found, meaning that the side reaction is more serious. So the evacuation valve design is also necessary for the practical application of the aqueous Zn-ion hybrid SC, in order to solve the safety issue, as same as the lead-acid battery or lead-carbon battery [41].

## 3. Materials and Methods

### 3.1. Materials Synthesis

The Polyaniline (PANI) nanofibers were prepared from aniline monomers by a rapidly-mix method as our previous works [24]. 9 mL aniline (AN, Sinopharm, Shanghai, China, >99.5%) was added to 300 mL 1 M dilute HCl, and 11.04 g ammonia peroxydisulfate (APS, Sinopharm, >98%) oxidant was dissolved into 300 mL 1 M dilute HCl. The AN solution was magnetically stirred for 0.5 h and cooled down to 0–5 °C, then the APS solution was added into the AN solution. The mixture was stirred at 0–5 °C for 24 h. The dark green precipitate was filtered under vacuum and washed with deionized water until the filtrate became colorless and neutral. The product was dried overnight at 80 °C. For the one-step carbonization and simultaneous chemical activation of PANI, the KOH and PANI were ground separately, mixed together with a weight ratio of KOH/PANI = 1:2, added alcohol as the grinding aid, and put into a nickel crucible. The crucible was placed into a horizontal steel tube and purged with Ar flow (100 mL min^−1^) to the activation temperature (650, 700, 750, 800, 850, or 900 °C) at a rate of 3 °C min^−1^. The activation process took 2 h, and then the sample was cooled in the furnace under Ar protection. The resulting sample was washed with 1 M HCl and deionized water until near-neutral (pH = 7) was reached before being dried in an oven at 80 °C for at least 24 h.

The FACs samples activated at 700 °C were heat treated at different temperatures (200, 400, 600, and 800 °C) for 2.5 h. These samples are denoted as FACs200, FACs400, FACs600, and FACs800 for short in the following sections. The carbon nanosponges (CNSs) sample was prepared by the KOH activation of carbonized PANI, as same as our previous work [4,5].

### 3.2. Material Characterization

The microstructure and morphology of the FACs were characterized by the scanning electron microscopy (SEM, JEOL, JSM-7001F, Tokyo, Japan) and transmission electron microscopy (TEM, FEI TS20 microscope, OR, USA). The specific surface area and pore size distribution (PSD) were obtained from N_2_ adsorption (−196 °C) isothermals performed on an (MicrotracBEL, BELSORP-MAX, Osaka, Japan) instrument with a relative pressure (P/P_0_) of 0.00000001 to 1. The samples were degassed at 200 °C for 12 h under turbo molecular vacuum pumping prior to the gas adsorption measurements. FTIR analyses of the functional groups were adopted by Nicolet iS50 (MA, USA). X-ray photoelectron spectroscopy (XPS) analyses were performed on a Thermo ESCALAB 250XI spectrometer, MA, USA).

### 3.3. Electrochemical Measurements

The positive electrode materials were prepared by milling the activated carbon powders (85 wt%) with 7 wt% Super-P carbon black and 8 wt% polyvinylidene fluoride binder (PVDF, Solvay, Shanghai, China) in 1-methyl-2-pyrrolidone (NMP) solvent, coating the mixture onto stainless steel foil, hot-rolling at 6 MPa, and cutting into a circular electrode with a diameter of 12 mm. The mass loading of the active material was approximately 2 mg cm^−2^. For practical application, an electrode with a thickness of 120 μm (mass loading of 6.7 mg cm^−2^) was also fabricated. The electrodes were dried in the vacuum oven at 120 °C overnight before assembling in argon-filled glovebox. Both the FACs positive and Zn foil (Sinopham) negative electrodes were assessed with 2025 coin cells, by using an air-laid paper as the separator. The electrolyte used was 2 M ZnSO_4_ (Sinopharm, >98%) in H_2_O. The cyclic voltammetry (CV) data were collected with a multi-channel potentiostat (PARSTAT MC, AMETEK, Oak Ridge, Tennessee, USA) at a scanning rate of 1–50 mV s^−1^ by using a 3-electrode cell (EL-CELL ECC-Ref Electrochemical Test Cell, Hamburg, Germany) with a tiny metallic Zn wire as the reference electrode, in order to eliminate the influence of the Zn metal negative electrode.

The gravimetric capacitance of the Zn-ion SCs was calculated from the galvanostatic discharge curve according to C = (IΔt)∙(mΔV)^−1^, where I is the constant discharge current, m is the mass of carbon materials on the electrodes, ΔV is the voltage change during the discharge process, and Δt is the duration of the discharge process.

### 3.4. Computational Details

The density functional theory (DFT) was used to calculate the geometry size of the cluster formed by SO_4_^2−^ and 12 H_2_O molecules (solvated SO_4_^2−^). Several kinds of the polymeric ion configurations were examined. All the calculations based on DFT were carried out using Vienna Ab initio Simulation Package code, with the projected-augmented-wave potential method. The model was built by cell parameters of a = b = c = 18 Å; α = β = γ = 90°.

The proton transition mechanisms on the FACs surface were demonstrated using density functional theory (DFT) calculations. All the calculations based on DFT were carried out using Vienna Ab initio Simulation Package (VASP) [42,43,44,45]. The generalized gradient approximation (GGA) in the form of the Perdew, Burke, and Ernzerhof (PBE) functional was used to approximate the exchange and the correlation. The FACs surface (graphene oxide) models were built by cell parameters of a = b = c = 18 Å; α = β = γ = 90°. The k-point meshes in the Brillouin zone (BZ) were sampled by 2 × 2 × 2. The convergence of plane-wave expansion was obtained with cut-off energy of 400 eV. Gaussian smearing with a width of 0.05 eV was used for the occupation of the electronic levels. Electronic self-consistent energy had a convergence accuracy of 1 × 10^−4^ eV. All structures were optimized until the forces on all unconstrained atoms were less than 0.02 eV Å^−1^.

The transition energy (*E_Tr_*) of the proton transfer was calculated as
*E_Tr_* = *E_tot_* – (*E_GOH_* + *E*_*SO*_4__)(2)
*E_Tr_* = *E_tot_* – (*E_GNH_* + *E*_*SO*_4__)(3)
where *E_tot_* is the total energy of the system after the proton transfer; *E_GOH_* is the energy of the functionalized graphene of the FACs with a –OH function; *E_GNH_* is the energy of the functionalized graphene of the FACs with a –NH function; and *E*_*SO*4_ is the energy of sulfate group. A negative *E_Tr_* value implies an energy favorable transition.

The charge density difference can be used to analyze the transfer of interatomic charge, and the charge accumulation regions and charge depletion regions are ascertained. The charge density difference was calculated by the equation as follows [46]:
(4)$$\Delta \rho =\rho_{tot}-\rho_{GO}-\rho_{H}-\rho_{{SO}_{4}}$$
(5)$$\Delta \rho =\rho_{tot}-\rho_{GN}-\rho_{H}-\rho_{{SO}_{4}}$$
where *ρ_tot_* is the total charge density of the functionalized graphene-proton-SO_4_ system; *ρ_GO_*, *ρ_GH_*, *ρ_H_*, and *ρ*_*SO*4_ are the charge densities of isolated graphene–O, graphene–N, proton, and sulfate group in the same combined structure.

## 4. Conclusions

Functionalized AC positive materials (FACs) were synthesized by the one-step carbonization and simultaneous KOH activation of polyaniline (PANI) nanofibers. The obtained FACs sample showed a micropores-rich structure. Zn-ion hybrid SCs based on FACs positive electrode and aqueous electrolyte revealed a significant improvement compared to the mesopores-riched carbon nanosponge materials (CNSs) and heat-treated FACs samples in the electrochemical performance. Especially, the change of capacitance is consistent with the change of C–NH and C–OH functions contents for all samples. The favorable micropores structure and the pseudocapacitance induced by the proton transfer between the C–OH, C–NH, and SO_4_^2−^ function lead to excellent electrode performance of FACs. The DFT simulation further proves the proton transfer mechanism. The FACs-based Zn-ion hybrid SCs exhibits a high capacitance of 435 F g^−1^, a high specific energy of 101 Wh kg^−1^ (at 67.8 W kg^−1^), high power energy of 14,100 W kg^−1^ (at 46.4 Wh kg^−1^) electrode materials. The estimated energy density of the full cell is approximately as high as 26.3 Wh L^−1^. 89% of capacitance retention was obtained after 10,000 cycles. Furthermore, high power density of 3666 W L^−1^ with 12 Wh L^−1^ was achieved. Moreover, the proton transfer effect can be further enhanced by introducing extra hydrogen ions in the electrolyte with low pH value. The highest capacitance of 544 F g^−1^ is obtained at pH = 3. The proton transfer process tends to take place preferentially on the C–OH rather than C–NH functions, based on the DFT calculation. However, the gas evolution was also found during cycling, meaning that the evacuation valve design is also necessary for the practical application of the Zn-ion hybrid SCs with aqueous electrolyte. The results would enlighten and promote the micropores-rich and hydrogen-containing group-rich activated carbon’ application. All of which put forward a new strategy of the research on the dual-ion energy storage devices.

## Figures and Tables

**Figure 1 molecules-24-02589-f001:**
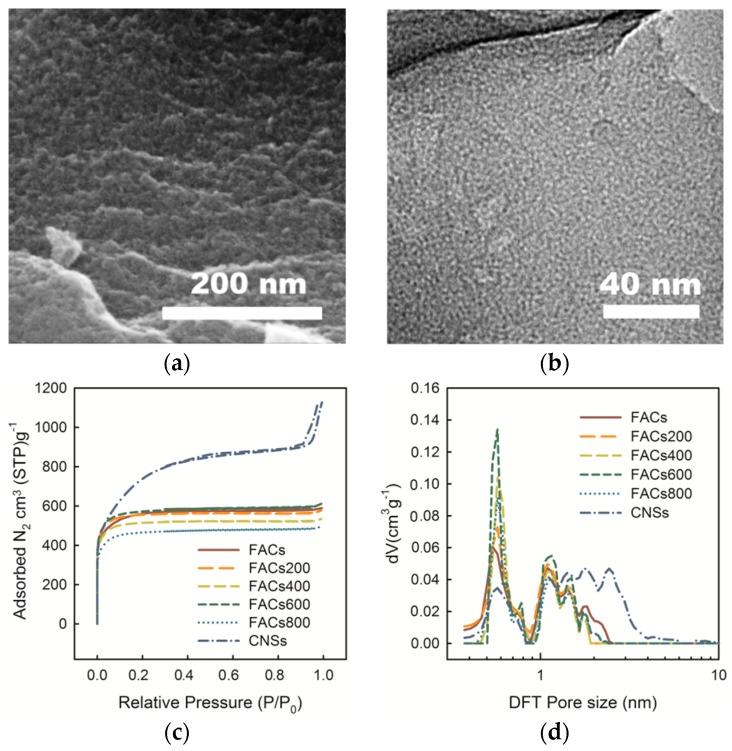
(**a**) SEM and TEM (**b**) images of the functionalized activated carbon cathode materials (FACs). (**c**) N_2_ isotherms and (**d**) pore size distributions (PSDs) of FACs, FACs200, FACs400, FACs600, FACs800, and carbon nanosponges (CNSs) calculated by nonlocal density functional theory (NLDFT).

**Figure 2 molecules-24-02589-f002:**
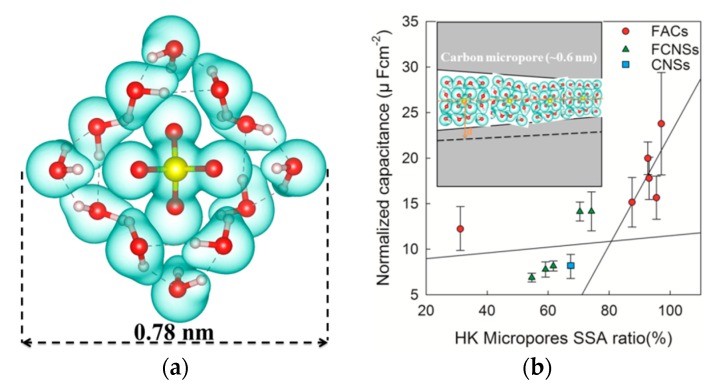
(**a**) Solvated configurations of the SO_4_^2−^·(H_2_O)_12_ cluster. Oxygen, hydrogen, and sulfur are represented in red, purple, and yellow, respectively. The dotted lines indicate hydrogen bonds. (**b**) Plot of specific capacitance normalized by BET specific surface areas (SSA) for the carbons in this study (FACs and CNSs) and in the other study (FCNSs) with identical electrolytes. The normalized capacitance increased with increasing of HK micropores SSA ratio with two-stage linear simulation. (**b** insert) Drawings of solvated ions residing in pores with distance between adjacent pore walls of ~0.6 nm illustrate this behavior schematically.

**Figure 3 molecules-24-02589-f003:**
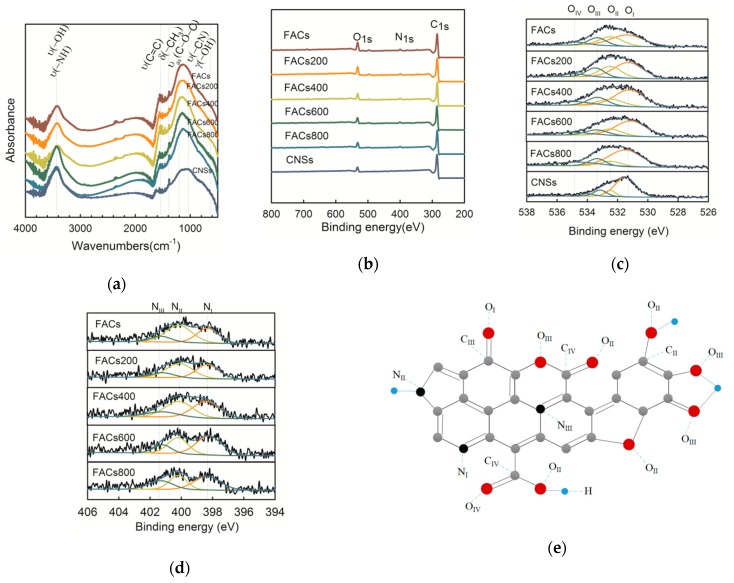
(**a**) FTIR spectra of six samples. υ, stretching; δ, bending (in-plane); γ, bending (out-of-plane); s, symmetric; as, asymmetric. (**b**) Experimental X-ray photoelectron spectroscopy (XPS) results of the six samples. (**c**) Comparison of the O1s deconvolutions for the six samples. (**d**) Comparison of the N1s deconvolutions for the five samples. (**e**) Schematic diagram of the FACs chemical structure.

**Figure 4 molecules-24-02589-f004:**
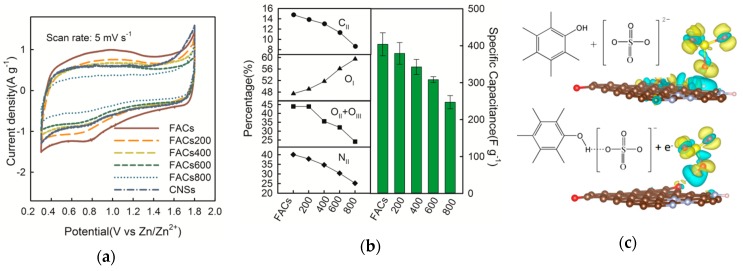
(**a**) Cyclic voltammetry (CV) curves of the six samples obtained using the carbon positive electrode as working electrode, and the pure Zn foil as both the counter and reference electrodes at 5 mV s^−1^ and 0.3 to 1.8 V vs. Zn/Zn^+^. (**b**) Comparison between the changes of functions contents and the change of the average capacitances at 0.1 A g^−1^ for the five samples in the Zn-ion hybrid supercapacitors (SCs) with aqueous electrolyte. DFT simulation of the proton transfer process on the –OH group (**c**) and –NH group (**d**) from the initial adsorption state to the final relaxed state. (**e**) Separation of the capacitive currents (*k*_1_*v*, blue shadow area) and diffusion currents (*k*_2_*v*^1/2^, blank area) for the FACs positive electrode in the Zn-ion hybrid SCs with aqueous electrolyte at the scan rate of 5 mV s^−1^. (**f**) The diffusion and capacitive contribution ratio in the total intercalated charge as a function of sweet rates during CV processes for three SCs samples with pH values of 3, 5, and 7.

**Figure 5 molecules-24-02589-f005:**
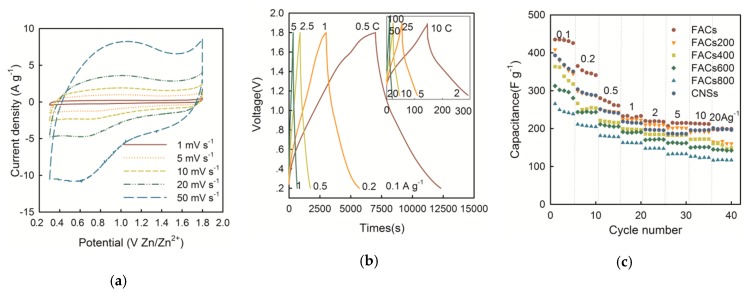
(**a**) CV curves of the FACs-based Zn-ion hybrid SC obtained using the carbon positive electrode as working electrode, and the pure Zn foil as both the counter and reference electrodes at various scan rates of 1–50 mV s^−1^ and 0.3 to 1.8 V vs. Zn/Zn^+^. (**b**) Galvanostatic charge/discharge (GCD) curves of FACs-based Zn-ion hybrid SC at low current densities of 0.1, 0.2, 0.5, and 1 A g^−1^ and (insert) high current densities of 2, 5, 10, 20 A g^−1^. (**c**) Rate capability: current density versus specific capacitance for the six samples. (**d**) Gravimetric Ragone plots of the Zn-ion hybrid SCs in aqueous electrolyte based on the mass of carbon positive electrode materials (FACs and CNSs) in this work, compared to the reported work by Dong et al. [18]. The estimated values of full device with thick electrode (120 μm) are also given. (**e**) Cycling stability of the FACs-based Zn-ion hybrid SC at 2 A g^−1^ for 10,000 cycles of charge-discharge.

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
