# Peer review of "Great Enhancement of Carbon Energy Storage through Narrow Pores and Hydrogen-Containing Functional Groups for Aqueous Zn-Ion Hybrid Supercapacitor"

_molecules, 2019, doi:10.3390/molecules24142589_

Round 1

Reviewer 1 Report

Dear Editor,

Here is the review with comments for the manuscript (molecules-543576)

Title: Great enhancement of carbon energy storage through narrow pores and 
hydrogen-containing functional groups for aqueous Zn-ion hybrid supercapacitor

In this manuscript, the author shared their recent investigation of Zn-ion hybrid supercapacitor by utilizing ZnSO4 as electrolyte, and functionalized activated carbon as cathode materials. Very high capacitance and very good stability has been achieved after PH modification. Further they explain such enhancement by DFT calculations.

Overall, this is a very important and interesting topic. The experimental are carefully designed. With the following modifications, I suggest a second review before recommendation for publication.

1)     In your abstract, aqueous is generally considered safe. However, Ionic liquid is also very safe with lager voltage window, therefore over voltage issue could also be eliminated. Please double check.

2)     In Fig 5e, what is your C rate if convert from A/g? Form the GC curve, it seems the columbic efficiency and charging efficiency would be quite low at lower rate, eg. 0.1 A/g.

3)     For your testing, do you need an electrochemical activation process for your performance/rate test?

4)     For the cell design, your composite demonstrated very good cycling. However, for large applications, the tapping density which in result for higher volumetric density and process ability is very critical, what is your case for carbon?

The following related papers should consider for reference

https://doi.org/10.1016/j.nanoen.2017.03.003

https://doi.org/10.1016/j.cej.2019.05.174

5)     For Fig 4 CV analysis, please put more supporting info for the cathodic and anodic contribution was from Zn/Zn2+.

6)     Following up the previous question, your voltage window from 0.3-1.8 V. It was actually higher than the decomposition potential of aqueous systems (1.25 V by thermodynamic). The peak at right end side is also very tilted. Please double check your efficiency.

It would be true that at very high rate, eg. like your case of 5mV/s the polarization of voltage shit would causes “fake” enlarged voltage.

7)     Fig 4e, how did you calculate your capacitive contribution? Depends on your composite with electrolyte conditions/systems. Sometimes the redox peak portion could have less capacitive contribution behavior (relatively sluggish). But also depends you’re your peak position during the analysis.

The following paper should be cited:

-        https://doi.org/10.1021/acsnano.6b08332

-        https://doi.org/10.1021/acs.jpclett.8b00200

8)     Why scientifically you think higher H+ concentration helps for capacity? Does that mean lowering PH could make de solvation energy lower? Be careful that carbon is a good catalysis for such low PH value. In other words, H2 bubble would be generated in your case of voltage window, right?

9)     What was the reason in Fig 5b got a sudden capacity drop at around 3000 cycles?

10) You have a very good value of specific energy. If based on total electrode mass, what would be your value?

11) The mass loading or areal capacity is meaningful for a performance report.

Reviewer 2 Report

In this manuscript, authors confirmed the proton transfer mechanism again in Zn ion hybrid super capacitor using carbon cathode materials. Figure 2b looks quite important, as the distortion is a key for capacitance enhancement. They also revealed the proton transfer process takes place on H-containing groups, which are the key factor for the electrochemical property, from XPS data as well as DFT calculation. Especially XPS deconvolution process was done very well for quantitative analysis. However, Figure 3a has to be modified for better view, and further IR analysis is required for the comparison with other carbon materials. Comments and questions are the following.

1. Which FAC is responsible for Figures 1a and 1b?

2. In Figures 1c and 1d, only line without symbols would be better. In addition, it is also recommended to change the color of CNS from pink to blue or other. 

3. In line 132, authors mentioned the FTIR spectra show the CNSs sample has fewer functions than the FACs samples. However, it isn't clear in Figure 3a. Y scale is too big to see the details of IR peaks. It is strongly recommended to decrease the distance between spectra for better view. Besides, several reference literatures are suggested for correct peak assignments.

4. The same colors could be convenient to the same samples, especially in Figures 1, 3, 4, and 5. 
